# Resonance Analysis and Gain Estimation Using CMA-Based Even Mode Combination Method for Flexible Wideband Antennas

**DOI:** 10.3390/s23115297

**Published:** 2023-06-02

**Authors:** Bashar Bahaa Qas Elias, Ping Jack Soh

**Affiliations:** 1Department of Communications Technology Engineering, College of Information Technology, Imam Ja’afar Al-Sadiq University, Baghdad 10052, Iraq; 2Centre for Wireless Communications (CWC), University of Oulu, P.O. Box 4500, 90014 Oulu, Finland; pingjack.soh@oulu.fi

**Keywords:** flexible antennas, wideband antennas, even mode combination, characteristic mode analysis

## Abstract

This work presents an efficient design and optimization method based on characteristic mode analysis (CMA) to predict the resonance and gain of wideband antennas made from flexible materials. Known as the even mode combination (EMC) method based on CMA, the forward gain is estimated based on the principle of summing the electric field magnitudes of the first even dominant modes of the antenna. To demonstrate its effectiveness, two compact, flexible planar monopole antennas designed on different materials and two different feeding methods are presented and analyzed. The first planar monopole is designed on Kapton polyimide substrate and fed using a coplanar waveguide to operate from 2 to 5.27 GHz (measured). On the other hand, the second antenna is designed on felt textile and fed using a microstrip line to operate from about 2.99 to 5.57 GHz (measured). Their frequencies are selected to ensure their relevance in operating across several important wireless frequency bands, such as 2.45 GHz, 3.6 GHz, 5.5 GHz, and 5.8 GHz. On the other hand, these antennas are also designed to enable competitive bandwidth and compactness relative to the recent literature. Comparison of the optimized gains and other performance parameters of both structures are in agreement with the optimized results from full wave simulations, which process is less resource-efficient and more iterative.

## 1. Introduction

Planar antennas are popular in biomedical applications due to their low profile, lightweight, low fabrication cost, and ease of integration with printed circuit board technology [1,2]. Recently, the evolution of the fifth generation (5G) of communication, the Internet of Things (IoT), big data applications, and vehicular communications [3,4] have facilitated communication with biomedical devices in therapy and medical diagnosis. These wireless-based assistive technologies are becoming more important in the continuous monitoring of patients’ daily lives in different situations. Wearable antennas should be compact, flexible, and have very low SAR values to be employed for human safety [5]. Flexible antennas are made from different conductive materials and substrates. The substrate is selected based on its dielectric properties, tolerance to mechanical deformations (bending, twisting, and wrapping), and endurance in external, often challenging environments. In contrast, the choice of conductive material determines the performance of the antenna, such as radiation efficiency.

In full-wave simulations, designers typically optimize a structure’s parameters based on their understanding of its operating principles. Generated wave behaviors and performance parameters are the main guides to determining the next optimization steps for antennas. This is aided by software using methods, such as Integral Equations (IE), which can be solved using the Method of Moments (MoM) for the complex environment with inhomogeneous lossy dielectrics, Finite Elements (FE), Finite Differences in the Time Domain (FDTD), and Finite Integration Technique (FIT). Such software-based antenna design and optimization processes are often iterative and depend heavily on the knowledge, experience, and understanding of designers on specific antenna topologies. This design process can easily be more tedious than expected, considering that more complex and non-arbitrary modern antennas of today are needed to meet growing requirements due to the demands of multi-standard communications. On the contrary, such a design process can be made more systematic and efficient by generating and understanding the physical insights of the structure using Characteristic mode analysis (CMA).

CMA is a method used in electromagnetics to provide insights into the intrinsic resonant qualities of a structure by locating and analyzing the structure’s basic modes [6,7,8,9]. This is because each physical structure has a collection of resonant modes, which are dependent on its structure, materials, boundary conditions, and frequency. The notion of CMA was first established in [10], then improved by [11,12]. The presence of the modes is independent of the excitation. However, different forms of excitations at distinct sites can be used to satisfy varied operational requirements. The use of the CMA enables the efficient tuning of an antenna to the correct resonant frequencies and determines the best locations to excite it. This is via the generation of the modes, modal significance, and surface current distributions and observing them over a frequency range of interest. Such a process will then significantly reduce the required calculation time and effort and at the same time, enable more targeted parameter adjustments within the optimization process.

In this work, a method to estimate gain over a wide operating bandwidth from antennas is proposed and applied to two flexible wideband antennas. Wide operating bandwidth is important for wearable antennas to ensure that they can overcome any performance deviation relative to the required frequency range due to the impact of surrounding human tissues [13]. The various active modes of a radiating structure were studied using the even mode combination (EMC) method. This is based on the principle of summing the electric field magnitudes of the first even dominant modes. To obtain the dominant modes, the modal significance of the structure is first observed at the desired frequency range using CMA. An overview of the EMC method using CMA is as follows:First, a radiating patch is designed without its ground, substrate, and excitation.Then, an analysis of the structure is performed using CMA to determine the even dominant modes. This is completed by observing the electric field (E-field) results throughout the target operating band.A suitable substrate is then added to the antenna prior to the excitation of the radiating patch. This antenna is analyzed to calculate the values of its forward gain.Estimation of the forward gain of the antenna takes place using the EMC method. The E-field magnitudes of the first two even modes are combined.Finally, the resulting maximum gain of this process provides an estimate of the highest gain value attainable over the frequency band of interest.

As the proposed method aims to improve the time- and resource-efficiency needed in the analysis and optimization effort of a complete antenna’s gain, results from EMC for these two wideband planar monopole antenna examples are compared with results from full wave simulations. Their design and main differences are as follows:Antenna 1: a planar monopole with a slotted radiator and fed using a coplanar waveguide (CPW). This antenna was designed on a flexible Kapton polyimide film with a thickness of 0.11 mm, whereas 0.01 mm-thick aluminum foil is used as its conductive sections (radiator, ground plane, and feedline).Antenna 2: a planar monopole with a spring-like radiator and fed using a microstrip line. The antenna is designed on textiles, with a 3 mm thick felt as the substrate and ShieldIt conductive textile (0.17 mm thickness) forming the conductive sections.

The rest of the paper is organized as follows. First, the topology, design procedure, and performance of both antennas in free space are explained in Section 2. Finally, Section 3 concludes this work.

## 2. Antenna Configuration

### 2.1. Antenna 1: Topology

This section presents the compact wideband antenna topology. The radiating patch consists of three main sections: a half-circle (at the bottom), ovoid (in the middle), and rectangular (at the top) patches, and connected with two rectangular arms on the left and right edges of the radiator. The CPW feeding technique is adopted in this work to create extra modes and minimize quality factor (Q), resulting in wide and multiband resonance behavior [14,15,16,17,18,19]. The antenna has an overall volume of 56 × 56 × 0.11 mm^3^ (0.69 *λ_g_* × 0.69 *λ_g_* × 1.37 *λ_g_*), printed on a Kapton polyimide film flexible substrate (relative permittivity of *εr* = 3.5, and a loss tangent of 0.002) with 0.11 mm thickness. The antenna has a partial ground plane with a width of 51.44 mm. FEKO simulator software is used to design and analyze the proposed antenna. Figure 1 shows the design of the Antenna 1. The main advantages of the proposed antenna design are as follows:The antenna has moderate gain and wide bandwidth in free space.Additionally, the fabricated antenna demonstrates good radiative properties when operating on both rigid and curved surfaces, making it suitable for different wearable communications applications.

**Figure 1 sensors-23-05297-f001:**
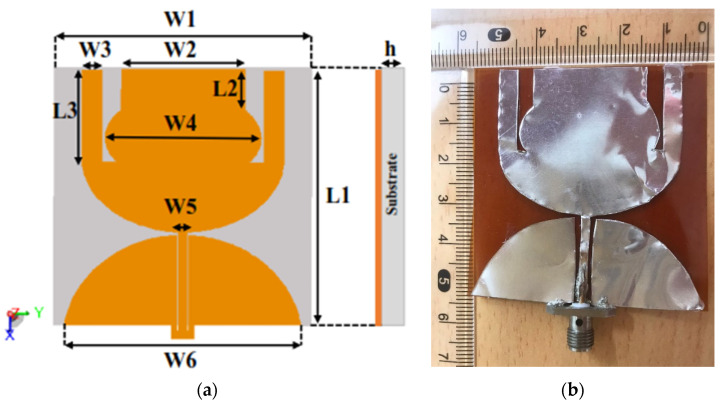
(**a**) Configuration of the proposed Antenna 1 with *L*1 = 56 mm, *W*1 = 56 mm, *L*2= 8.922 mm, *L*3 = 20 mm, *W*2 = 27 mm, *W*3 = 4.5 mm, *W*4 = 33.92 mm, *W*5 = 2 mm, *W*6 = 51.44 mm, *h* = 0.11 mm, (**b**) fabricated prototype of Antenna 1.

#### 2.1.1. Antenna 1: Design Procedure

The proposed Antenna 1 is designed in four steps, as seen in Figure 2a–d. Different parasitic patches are integrated with the main radiator to improve its performance. The simulation process using CMA includes only the radiating patch as a perfect electrical conductor (PEC) without the substrate, ground, and excitation. The modal significance (MS) parameter, which ranges between 0 < MS ≤ 1 is observed in CMA simulations. Modes with values of MS greater than 0.707 are active modes and have high potential to be excited using the specific structure shown in Figure 2. Then, the MS results in each step of design are compared with the reflection coefficient of the complete antenna (with substrate, ground, and excitation) based on the normal method of moments (MoM) (see Figure 3a–d). The simulation process using CMA provides the natural resonance of the modes, which can help in understanding the antenna operation and estimating the antenna bandwidth based on physical insights from MS in less time.

The steps involved in the antenna design can be summarized as follows:As the first step, a conventional circular patch antenna is designed. In this iteration, the antenna is not matched at the resonant frequencies from 2 to 2.2 GHz. Similarly, no modes are active to cover this range of frequencies, as illustrated in the MS plot in Figure 3a.Next, the antenna in the previous step is modified in step 2. A portion from the upper part of the circular patch is removed to achieve miniaturization at the same resonance, as shown in Figure 3b.The antenna is further modified by adding two rectangular parasitic patches on the left and right sides of the radiator. It is observed that the antenna operated with a 10-dB reflection coefficient starting from 2 GHz, which is generated by mode 1, as in Figure 3c. However, the reflection coefficient in this step shows a narrowband around 3 GHz, which does not meet the targeted wideband antenna operation.To overcome this, the top side of the patch is extended by connecting a rectangular-shaped resonator. Using this step, the proposed antenna achieved a bandwidth from 2 GHz to 6 GHz, as shown in Figure 3d. Some differences between the estimated operating bandwidth using the MoM and CMA approaches originates from the effect of the excitation process in generating the excited modes.

Additionally, to explain the antenna radiation mechanism, the surface current distributions based on CMA on the patch are shown in Figure 4. In the first mode, the current is concentrated near the beginning of the two parasitic arms connected with the left and right sides of the patch at 2.45 GHz and 3.6 GHz. In comparison, the current distribution is extended to include the upper and lower sides of the patch with varying intensities at 5.5 GHz and 5.8 GHz. In mode 2, it is significantly observed that the current flow in the feed line is at 2.45 GHz, 5.5 GHz, and 5.8 GHz, respectively. Here, the current still propagates along the left and right portions of the radiating patch, except at 2.45 GHz.

#### 2.1.2. Results and Discussions

Antenna Performance in Free Space

This section presents the results and analysis of the proposed wideband antenna in free space in terms of reflection coefficient, radiation patterns, surface current distributions, efficiency, and gain. First, Figure 5 shows the simulated and measured reflection coefficient of the antenna on the Kapton polyimide film substrate. Simulations showed that the proposed antenna spans a frequency band larger than 4 GHz with an operational range from 2 to 6 GHz, with satisfactory impedance matching. The reflection coefficient reaches a minimum value of −32.21 dB at roughly 2.7 GHz. On the other hand, the measured reflection coefficient indicates that the antenna has a very good impedance matching with a reflection coefficient below −10 dB from 2 to 5.27 GHz. This shows that the antenna has a bandwidth larger than 3.2 GHz.

Figure 6a,b illustrates the simulated polar radiation patterns and the co- and cross-polarization of the proposed antenna, respectively. The far-field patterns over a range of frequencies indicate that the antenna featured a quasi-omnidirectional pattern in the major H-plane (*θ* = 90°), while a bidirectional radiation pattern is found for the E-plane (*θ* = 0°). As the frequency increases, the lower half of the pattern shows a significant distortion with more ripples. Additionally, it is found that the radiation is reduced from the non-radiating edges that are responsible for cross-polarized radiation, which is mainly caused by the currents on the feeding probes. The designed antenna exhibits a peak gain of 8 dBi at 3.6 GHz, 5 dBi at 2.45 GHz, and 4 dBi at 5.5 GHz and 5.8 GHz, respectively.

b.Gain Estimation using EMC Method based CMA

The first design example (Antenna 1) depicting the efficiency of this method is presented and explained in detail as follows:i.Based on characteristic mode (CM) theory, the proposed Antenna 1 is first analyzed with only the perfect conductor (patch) and without any substrate, ground, and excitation to determine the natural modes of resonance.ii.After the simulation process is performed, the electric field of the first two even modes (mode 2 and mode 4) is obtained from the results, as illustrated in Figure 7.iii.Next, these two modes are then combined to form a wide specific band (from 2 to 6 GHz) as represented in Equation (1). For example, at 3.7 GHz, the electric field of mode 2 is 5.79 V, whereas it is 6.62 V at mode 4. Then, the EMC method is applied as follows:
EMC (V) = Em2 (V) + Em4 (V) (1)
EMC (V) = 5.79 V + 6.62 V
EMC (V) = 12.42 V
Which represents the maximum value of the EMC achieved at 3.7 GHz, as shown in the red box in Figure 7.

iv.The same procedure is performed for the rest of the frequencies. In Table 1, a set of frequencies is selected to verify the proposed method. The results of EMC in the fourth column are arranged in descending order (from the highest value of the electric field to the lowest), as also seen in Figure 8.

**Figure 7 sensors-23-05297-f007:**
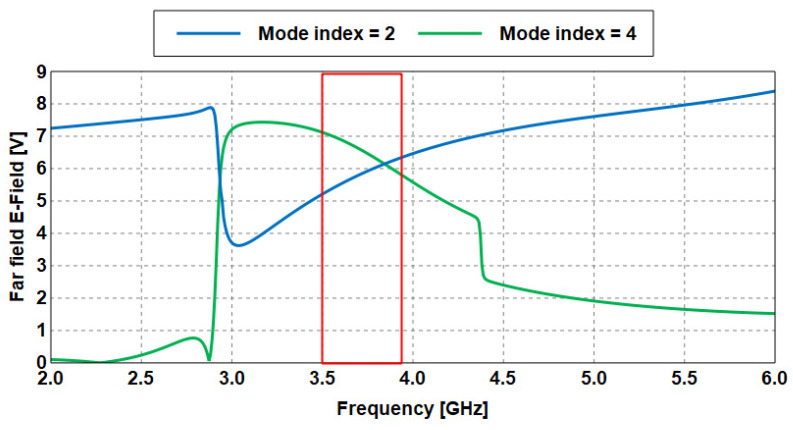
Electric fields of the wideband antenna based on CMA.

**Figure 8 sensors-23-05297-f008:**
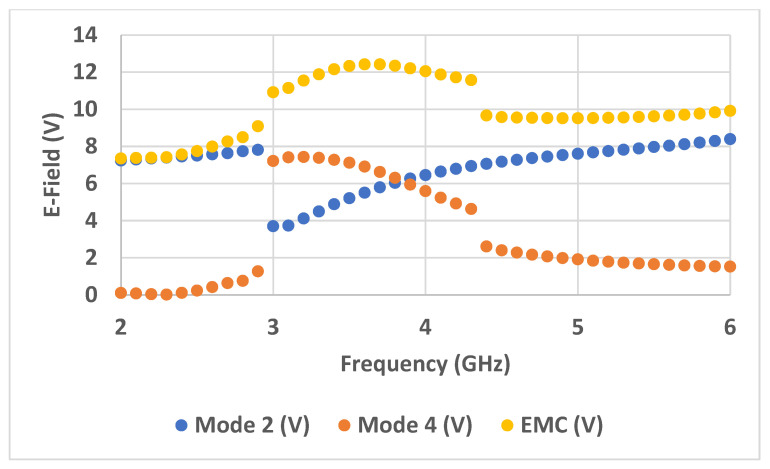
EMC results of the wideband Antenna 1 based on CMA.

v.Finally, the radiating patch is integrated with the substrate and excitation to form the complete antenna. Analysis indicated that the maximum gain of 3.54 dBi is achieved at 3.83 GHz, as indicated in the red box in Figure 9. Note that the frequency at which a maximum electric field was achieved based on the EMC-based CMA method agrees closely with the frequency of the maximum gain of the overall antenna.

### 2.2. Antenna 2: Topology

For further verification of the proposed method, an additional flexible patch Antenna 2, also referred to as “spring antenna” is used. This antenna featured a compact dimension of 32 × 42 × 3 mm^3^ (0.38 *λg* × 0.5 *λg* × 0.036 *λg*) and operated from 2.99 GHz to 5.57 GHz with a bandwidth of 73.8% [20]. Figure 10 illustrates the topology of Antenna 2 designed on a felt textile substrate with a thickness (*h*) of 3 mm, a dielectric constant of approximately 1.3, and a loss tangent of 0.044. The single-layered substrate ensures a low-profile structure and reduces potential complexities in fabrication and integration with clothing. The top patch and partial bottom ground are made using a ShieldIt conductive fabric. The overall antenna is fed by a microstrip feed line, with its width (*Wf*) optimized to ensure good impedance matching.

The steps to calculate the EMC values of Antenna 1 is then applied to Antenna 2 to further validate the proposed method. Only its radiator is first analyzed using CMA. Figure 11 shows the electric field of modes 2 and 4, which are then combined, as seen in Figure 12. From the results obtained in Table 2, it is observed that the four maximum values of the EMC of 10.22, 9.99, 9.72, and 9.46 V are achieved at 6, 5.9, 5.8, and 5.7 GHz, respectively, as also indicates in the red box in Figure 11.

The gains produced from the overall antenna indicate that the maximum gain is achieved at 5.77 GHz, as illustrated in the red box in Figure 13. This also agrees with the frequency where the maximum gain is obtained using the EMC-based CMA method.

Several studies in the literature have proposed approaches to estimate the performance of antennas based on different equations and techniques, as summarized in Table 3. Most of these works depend on the analysis of the complete antenna (i.e., with substrate, ground, and excitation) or are derived from measurement data. Additionally, artificial intelligence-based approaches in the literature, such as machine learning (ML) and deep learning (DL) methods, also provide an accurate estimation of the performance of antennas. However, such methods require extensive knowledge of the application of such algorithms, and may potentially incur additional memory and processing costs depending on the structure complexity, number of parameters for optimization, and search range. In comparison, the proposed method enables an efficient, simple-to-understand, and reasonably accurate procedure to predict the maximum gain of the antenna. Being only dependent on the radiator structure without the effect of substrate and excitation, this also minimizes the needed time in the design and analysis process.

A detailed comparison of the proposed antenna with other state-of-the-art flexible planar antennas from the literature operating in the same frequency range is summarized in Table 4. It can be observed that the proposed antenna featured an enhanced gain while being competitive in terms of compactness and bandwidth, as follows:

a.The size of the antenna is more compact compared to antennas proposed in [32,34,35,37,38].b.Its impedance bandwidth is improved compared to antennas proposed in [32,33,34,35,36,37,38,39,40].c.Higher gains are achieved compared with most of the reported antennas in the literature at the relevant frequency (considering the equivalent electrical sizes).

## 3. Conclusions

This work describes an efficient design and optimization approach to improving the gain of a wearable antenna based on the CMA approach. Prediction of the forward gain is proposed based on the even mode combination method, which is derived based on the characteristic mode analysis. To predict the forward gain, the combination in the electric field of the first two even modes is computed. Two wearable antennas made using different flexible materials and fed using different feed types are used to validate this method. Both antennas are analyzed over ranges of frequencies from 2 to 6 GHz. Applying the EMC method, it can be observed that the maximum value of gain is achieved between 3.5 and 3.9 GHz, which is in good agreement with optimized values from full wave simulations (with substrate and excitation). Similarly, EMC analysis for the second wearable antenna designed on textile and fed using a microstrip line also enabled an accurate estimation of its maximum gain between 5.7 and 6 GHz. The simulated performance of the designed antenna is analyzed in terms of its reflection coefficient, radiation pattern, gain, and surface current. Both antennas are fabricated, and measurement results showed an acceptable agreement with simulations in terms of reflection coefficients and bandwidth. The first antenna operated from 2 to 5.27 GHz (measured) with gains of 8 dBi at 3.6 GHz, whereas the second antenna operated from 2.99 to 5.57 GHz (measured) and gain of 6 dBi at 3.5 GHz. Their operation in several important wireless frequencies, such as 2.45 GHz, 3.6 GHz, 5.5 GHz, and 5.8 GHz, and their flexibility enable them to be potentially applicable for WBANs, 5G, and IoT communications. The use of such a CMA-based approach can be extended in future works to include new methods to estimate the backward radiation of wearable antennas integrated with the artificial magnetic conductor (AMC). This will then also enable a more efficient prediction of the FBR parameter with less time cost by utilizing the features offered by CMA.

## Figures and Tables

**Figure 2 sensors-23-05297-f002:**
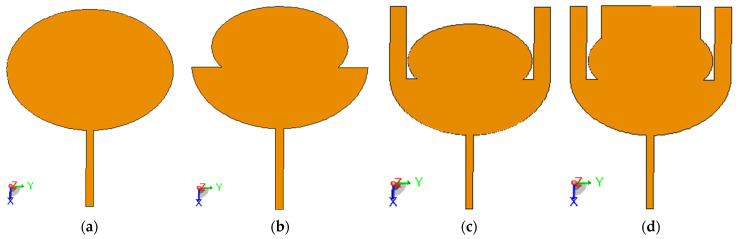
Design procedure of the proposed Antenna 1 (**a**) step 1, (**b**) step 2, (**c**) step 3, (**d**) step 4.

**Figure 3 sensors-23-05297-f003:**
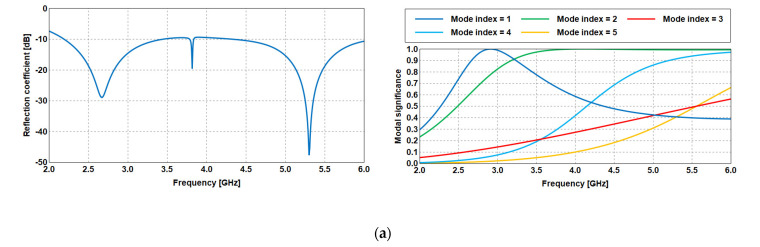
Design steps: reflection coefficient (**left**), modal significance (**right**). (**a**) Step 1; (**b**) Step 2; (**c**) Step 3; (**d**) Step 4.

**Figure 4 sensors-23-05297-f004:**
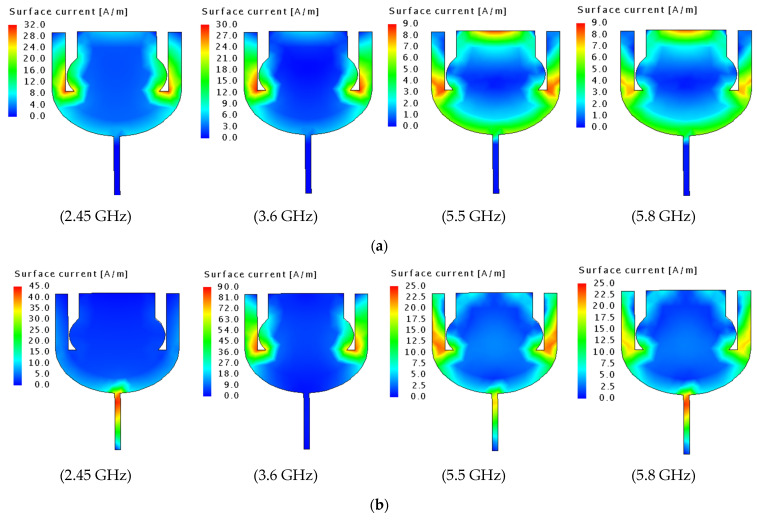
Surface current distribution using CMA (**a**) mode 1 (**b**) mode 2.

**Figure 5 sensors-23-05297-f005:**
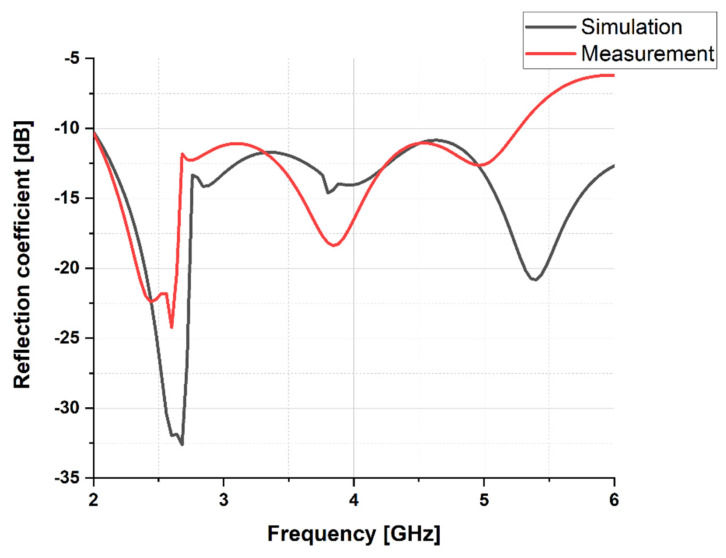
Simulated and measured reflection coefficients of the designed antenna.

**Figure 6 sensors-23-05297-f006:**
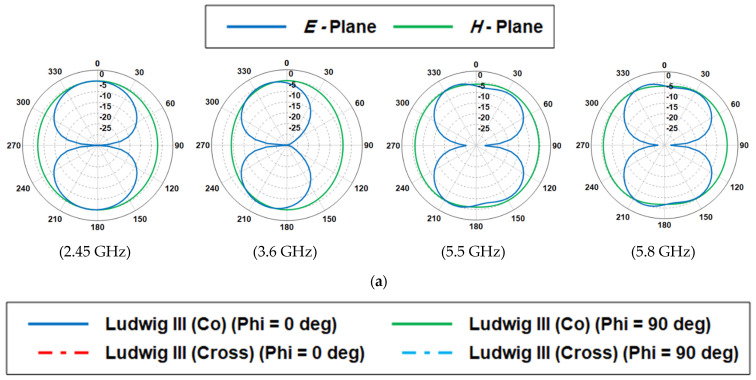
Simulated far-field pattern of the proposed antenna (**a**) *E*-plane and *H*-plane, (**b**) Co-pol and Cross-Pol.

**Figure 9 sensors-23-05297-f009:**
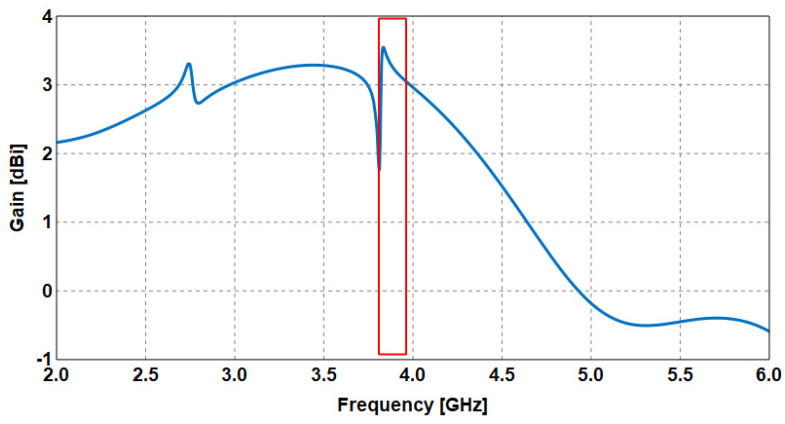
Gain of the wideband antenna.

**Figure 10 sensors-23-05297-f010:**
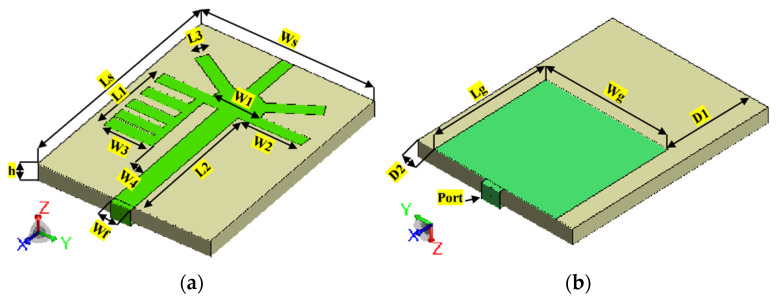
Configuration of Antenna 2 from [20] (**a**) Front view, (**b**) Back view. *Ls* = 42 mm, *Ws* = 32 mm, *L*1 = 15.3 mm, *L*2 = 26.625 mm, *L*3 = 2.25 mm, *W*1 = 8.97 mm, *W*2 = 11.625 mm, *W*3 = 9 mm, *W*4 = 2.625 mm, *Wf* = 3.75 mm, *h* = 3 mm, *Lg* = 24 mm, *Wg* = 25 mm, *D*1 = 18 mm, and *D*2 = 3.5 mm.

**Figure 11 sensors-23-05297-f011:**
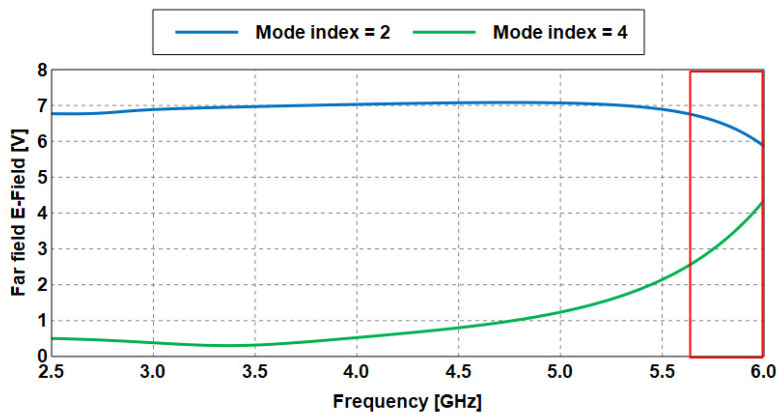
Electric field of the proposed spring antenna based on CMA.

**Figure 12 sensors-23-05297-f012:**
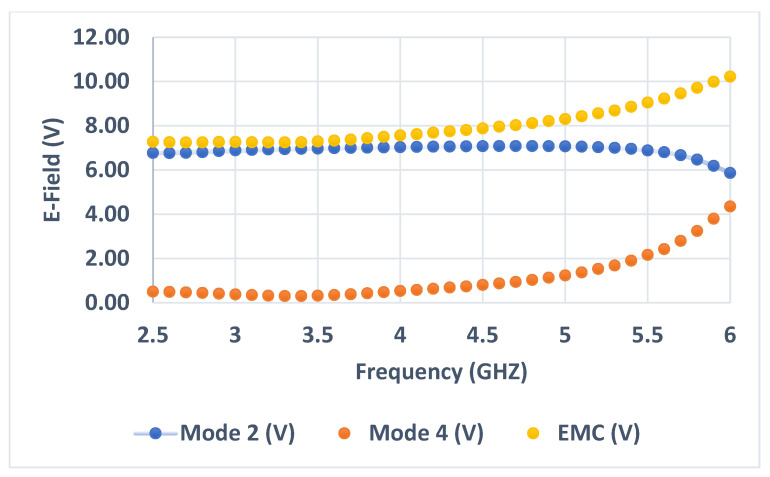
EMC results of the proposed spring antenna based on CMA.

**Figure 13 sensors-23-05297-f013:**
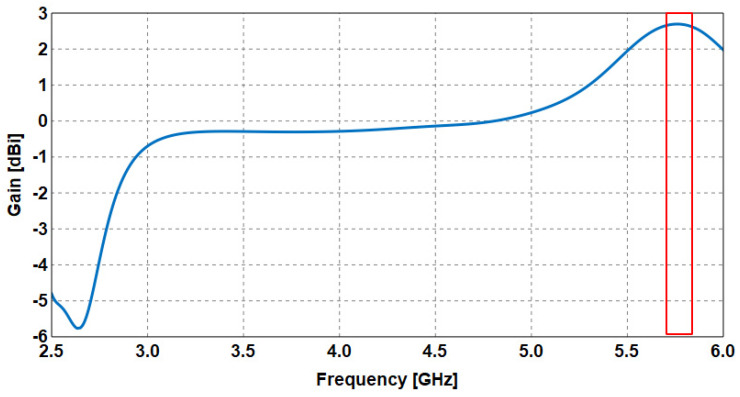
Gain of the proposed spring antenna.

**Table 1 sensors-23-05297-t001:** EMC results of a wideband Antenna 1 based on CMA.

Frequency (GHz)	Mode 2 (V)	Mode 4 (V)	EMC (V)
3.7	5.79	6.62	12.42
3.6	5.51	6.91	12.42
3.8	6.04	6.31	12.34
3.5	5.21	7.12	12.33
3.9	6.27	5.94	12.21
3.4	4.88	7.28	12.16
4	6.46	5.59	12.05
3.3	4.49	7.38	11.88
4.1	6.64	5.23	11.87
4.2	6.79	4.93	11.72
4.3	6.93	4.63	11.57
3.2	4.12	7.43	11.55
3.1	3.74	7.41	11.15
3	3.69	7.22	10.92
6	8.39	1.52	9.92
5.9	8.29	1.54	9.83
5.8	8.21	1.56	9.76
5.7	8.12	1.59	9.71
4.4	7.05	2.61	9.67
5.6	8.03	1.62	9.66
5.5	7.97	1.65	9.62
5.4	7.89	1.69	9.58
4.5	7.18	2.41	9.58
5.3	7.82	1.74	9.56
4.6	7.27	2.28	9.55
5.2	7.75	1.79	9.54
4.7	7.37	2.17	9.54
5.1	7.68	1.85	9.53
4.8	7.45	2.07	9.53
5	7.61	1.91	9.52
4.9	7.53	1.98	9.52
2.9	7.82	1.27	9.08
2.8	7.73	0.76	8.49
2.7	7.63	0.64	8.27
2.6	7.56	0.42	7.99
2.5	7.51	0.24	7.74
2.4	7.45	0.11	7.57
2.3	7.39	0.02	7.42
2.2	7.34	0.04	7.39
2.1	7.29	0.08	7.37
2	7.24	0.11	7.35

**Table 2 sensors-23-05297-t002:** EMC results of the proposed spring antenna based on CMA.

Frequency (GHz)	Mode 2 (V)	Mode 4 (V)	EMC (V)
6	5.86	4.35	10.22
5.9	6.19	3.80	9.99
5.8	6.47	3.25	9.72
5.7	6.67	2.79	9.46
5.6	6.81	2.42	9.23
5.5	6.89	2.16	9.05
5.4	6.96	1.90	8.86
5.3	7.00	1.68	8.69
5.2	7.03	1.53	8.56
5.1	7.06	1.37	8.43
5	7.07	1.23	8.30
4.9	7.08	1.14	8.21
4.8	7.08	1.03	8.11
4.7	7.08	0.94	8.03
4.6	7.08	0.88	7.96
4.5	7.08	0.80	7.88
4.4	7.07	0.74	7.81
4.3	7.06	0.69	7.75
4.2	7.05	0.63	7.68
4.1	7.04	0.58	7.62
4	7.03	0.53	7.57
3.9	7.02	0.48	7.50
3.8	7.01	0.43	7.44
3.7	6.99	0.38	7.38
3.6	6.98	0.35	7.33
3.5	6.97	0.32	7.29
2.5	6.77	0.50	7.27
3	6.89	0.38	7.27
2.9	6.85	0.41	7.27
3.4	6.96	0.30	7.26
3.1	6.91	0.35	7.26
2.6	6.77	0.49	7.25
2.8	6.81	0.44	7.25
3.2	6.93	0.32	7.25
3.3	6.94	0.31	7.25
2.7	6.78	0.47	7.24

**Table 3 sensors-23-05297-t003:** Comparison of the proposed method with other approaches in the literature.

Ref	Study	Method Used	Advantages
[21]	Gain characteristics estimation of heteromorphic RFID antennas using neurospace mapping.	Gain estimation method for heteromorphic RF identification (RFID) antennas, built on the basis of neuro-SM technique.	Effectively reduces time consumption and avoids laborsome measurement processes.
[22]	Estimation of gain enhancement replacing PTFE by air substrate in a microstrip patch antenna.	Replacing PTFE by air substrate.	Very handy and useful to a designer dealing with air-substrate microstrip antennas.
[23]	Accurate antenna gain estimation using the two-antenna method.	Two-antenna gain determination method using Friis equation.	Eliminates the need to perform the measurement at various separation distances.
[24]	Contactless gain pattern estimation from backscattering coefficient measurement performed within a reverberation chamber.	Evaluated for the first time from the backscattering coefficient measured within a reverberation chamber.	The antenna backscattering coefficient is measured for two load conditions, namely, an open circuit and a 50 load. This distinguishes the radiation and structural modes from the total backscattered field.
[25]	A principal-component approach to antenna impedance estimation at MISO receivers.	Study the antenna impedance estimation at MISO receivers over Rayleigh fading channels and derive the optimal ML estimator in closed-form.	Numerical results suggest a computationally efficient, principal-components approach that estimates antenna impedance in real-time and shows sizable improvement against a reference estimator at low SNR.
[26]	Estimation of the axial ratio of the single-port dual-mode circularly polarized antenna using reflection coefficient.	The estimation method applies to a single-port CP antenna that is composed of two resonant modes with orthogonal polarization and 90° phase difference.	Simple way to can estimate the axial ratio (AR) of a circularly polarized (CP) antenna Illustrates the source of estimation errors and proves that the method reaches acceptable AR estimation without time-consuming microwave chamber measurement Provides a new perspective to high-speed pre-test and tuning for volume production of CP antennas.
[27]	Biased estimation of antenna radiation efficiency within reverberation chambers due to the unstirred field: role of antenna stirring.	Four measurement campaigns were successively performed: without antenna stirring, with platform stirring only, with source stirring only, and with a combined stirring of both antennas.	An extensive study on the role of antenna stirring in the retrieved radiation efficiency. Performing only source stirring or platform stirring leads to a biased estimation of the radiation efficiency.
[28]	Efficient yield estimation of multiband patch antennas using NLPLS-based PCE.	The tolerances associated with the fabrication process are applied to the statistically significant system parameters, and a Monte Carlo (MC) simulation is completed to accurately estimate the yield.	NLPLS-based PCE effectively reduces the system dimensionality using NLPLS and simultaneously extracts the statistical information on the same sample set (yield) using PCE. A solution for complex antenna yields analysis.
[29]	Modelling of dielectric resonator based filtenna for 2.5/2.6 (B41/n41) bands using machine learning algorithms.	Different parameters of the proposed antenna, such as reflection coefficient and gain, are optimized and predicted using different machine learning (ML) techniques, i.e., deep neural network (DNN), random forest, and XG boost.	An optimized value obtained from various ML algorithms is compared with the value obtained HFSS EM simulator and experimental result. Good agreement is obtained among all of them.
[30]	Data-driven surrogate modeling of horn antennas for optimal determination of radiation pattern and size using deep learning.	The geometrical design variables, operation frequency, and radiation direction of the design taken as the input, while the realized gain of the design is taken as the output of the surrogate model. Series of powerful and commonly used artificial intelligence algorithms, including deep learning, had been used to create a data-driven surrogate model.	Achieve a computationally efficient design optimization process for horn antennas with high radiation performance, besides being small in or within the limits of the desired application. A total of 80% computational cost reduction was obtained via the proposed approach.
This work	Resonance analysis and gain estimation of a wideband antenna using CMA.	EMC method based on characteristic mode analysis.	Estimates the gain of the antenna without using substrate, ground, and excitation, offering efficient time cost in the simulation process.

**Table 4 sensors-23-05297-t004:** Performance comparison of the proposed antenna with other flexible antennas available in the literature (with *λg* = lower operating wavelength).

Ref	Size of the Antenna (*λ_g_*)	Substrate Material	Operating Frequency (GHz)	Bandwidth (%)	Gain (dBi)	Flexible?	Year
[31]	0.27 × 0.2 × 0.003	Rogers 5880	3.5/6.8/9	112	>2.5	Yes	2023
[32]	0.83 × 0.83 × 0.073	Felt	3.5/5.8	11.7/9.1	6.6/7.2	Yes	2022
[33]	0.64 × 0.64 × 0.0125	Denim	2.45/3.32/ 3.93/5.8	3.7/5.7/ 5.85/9.8	−0.81/−2.81/−1.16/2.83	Yes	2022
[34]	0.83 × 0.83 × 0.053	Felt	3.5/5.8	12.5/11.6	6/5.8	Yes	2021
[35]	0.89 × 0.89 × 0.018	Felt	2.45/5.8	4.9/3.8	5.93/6.02	Yes	2020
[36]	0.63 × 0.63 × 0.018	Felt	2.38/3.5	4.1/7.0	1.38/7.7	Yes	2020
[37]	1.3 × 0.74 × 0.059	Felt	0.433/2.45	8.1/12.6	−0.5/6.1	Yes	2017
[38]	0.79 × 0.68 × 0.041	PDMS	2.43/5.15	8.2/17.5	2.2/3.02	Yes	2019
[39]	0.3 × 0.21 × 0.0006	conductive polymer (PANI/MWCNTs)	2.2/5.3	45/60	3.1 dBi at 5.8 GHz	Yes	2018
[40]	0.48 × 0.48π × 0.041	PDMS	2.45/5.8	3.3/4.17	4.16/4.34	Yes	2017
Prop.	0.69 × 0.69 × 0.001	Kapton polyimide film	2.45/3.6/5.5/5.8 Wideband (2–6 GHz)	100	5/8/4/4	Yes	2023

## Data Availability

Not applicable.

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
