# Peer review of "Resonance Analysis and Gain Estimation Using CMA-Based Even Mode Combination Method for Flexible Wideband Antennas"

_sensors, 2023, doi:10.3390/s23115297_

Round 1

Reviewer 1 Report

This manuscript claims to present "Resonance Analysis and Gain Estimation using CMA-Based Even Mode Combination Method for Flexible Wideband Antenna". However, I do not support its publication in its current form and much work is required to improve this paper.

My comments on this work are as follows:

Point1. The abstract did not provide any contribution to the manuscript. It should contain some kind of novelty. There are various similar designs which are much more compact and have better performance than this, published in the literature. Provide a clear contribution in the abstract.

Point 2: The introduction is long and contains irrelevant materials. The authors have not discussed anything about the applications and similar designs. It should be short, with relevant designs and applications.

Point 3: Such designs are mostly used for some applications, for example, collision avoidance for the blind, but the authors have not discussed such applications in detail in the paper (except on page 3, line 3). Please clearly explain them in the whole paper.

Point 4: Plot the input impedances for all these topologies and show that the imaginary part of the input impedance (Xin) goes through zero at the designed frequency.

Point 5:  In Figure 8, on page 8, the simulated radiation pattern is suspicious. Provide the on-body chest measurement radiation inside the anechoic chamber and need clearly explanation.

Point 6: The authors must provide the results for the proposed design in terms of bending around different cylindrical arms, such as 50 mm, 60 mm, 70 mm, 80 mm, 90 mm, and 100 mm, stretching, and humidity, which are highly important in the wearable antennas.

Point 7: Provide the simulated or measured result for the SAR and describe it more in detail, e.g., for which frequency range they are designed with standard dimensions. Further, discuss the distance gap between the antenna and the body.

Point 8: Plot a graph in dB for the (S21) S-parameters in LOS and NLOS for on-body locations.

Point 9: It will be great if the manuscript is revised for english corrections and the lengthy sentences are reduced.

Reviewer 2 Report

The paper is not well written. The flow of the paper is not clear. First, the authors provide a very extensive description of their design method for a planar monopole antenna. The antenna has no novelty and there are many similar antennas in the literature. There are too many figures and tables to show the CMA method, but there is no radiation pattern measurement. The performance of the antenna on the body and on flat and curved surfaces should be characterized via simulation and measurement. It is not clear if the gain results are from simulation or measurement, in flat or bent conditions, in free space, or on a human foot. No SAR analysis is provided. 

Then suddenly there is a new antenna shown in Fig. 12 without any relation to the first antenna. In the introduction, a 3rd antenna is shown in Fig. 2 from [16], that is not clear if the authors have permission to copy the figure here. If the authors would like to show their antenna is applicable in such a study they need to do many more measurements on various subjects and analyze the results.

In my opinion, the paper does not have a good organization, does not show any novelty, and is not suitable for publication. I recommend focusing on the novel aspect of the work which seems is the use of CMA for the design of the antenna, and reduce the length, and providing a better organization for the paper. If the application in the wearable conditions is considered information about the performance of the body model in simulation and measurements must be provided. Also, the SAR analysis should be added.

Reviewer 3 Report

An interesting topic is selected by the authors. With respect to the wide range of its applications and the need for high performance antenna designs for modern applications, performance analyses of the microwave antennas is a hot topic that being persuaded by many researchers.

Authors had proposed a work that present an approach based on characteristic mode analysis (CMA) to predict the resonance and gain of wideband antennas made from flexible materials. Ta detailed analyses of the proposed approach is explained by the authors. The overall presentation is seems ok but can be improved. However the most problematic thing about work is that the novelty and contribution of the work to the literature is not clearly presented. Although authors have presented a table of comparison of counterpart methods form literature in table 3, this table need to extend with more examples from state of the art methods. What about usage of artificial intelligence based surrogate models for prediction performance measures of antenna such as scattering parameters [A1] and or radiation pattern [A2]? With respect to the well-known application and potential of artificial intelligence what would be your approaches advantages and disadvantages?

[A1] Sharma, Rishi, Anjali Potnis, and Vijayshri Chourasia. "Modelling of Dielectric Resonator based Filtenna covering 2.5/2.6 (B41/n41) Bands using Machine Learning Algorithms." AEU-International Journal of Electronics and Communications (2023): 154642.

[A2] Piltan, OCKizilay, ABelen, MAMahouti, PData driven surrogate modeling of horn antennas for optimal determination of radiation pattern and size using deep learningMicrow Opt Technol Lett202319doi:10.1002/mop.33702

The literature overview of the work must be extended for such a challenging and highly persuaded work there are series of recently published works such as [A1] and [A2]. Author must extend their literature overview and table of comparisons with more cited works from 2023 and 2022.

Other suggestion are

-          Some of the figures are blurry, improve the overall quality of figures,

-          Table 1, is it really need to give 5 digits after “.” Instead of 5.79689 it would be better to round the value as 5.79 or 5.8 or if this value is extremely sensitive authors must explain why such sensitivity is a must.

-          The conclusion section can be extended, current work draw backs or not studied material alongside of a brief explanation about future works

Round 2

Reviewer 1 Report

Thank you so much that the authors have worked on my comments. However, the comments below still need to be addressed to improve the quality of the manuscript.

Point 1:  Concern # 1, from previous points is not addressed properly. The abstract does not provide any contribution to the manuscript. A similar design has been published in the literature [1]. Please provide a clear contribution to the design structure as well as in the abstract.

[1] S. Jun, B. Sanz-Izquierdo and M. Summerfield, "UWB antenna on 3D printed flexible substrate and foot phantom," 2015 Loughborough Antennas & Propagation Conference (LAPC), Loughborough, UK, 2015, pp. 1-5.

Point 2: Concern # 6, is not properly addressed (Figure 20 is not correct).  The authors must show it physically bending around the cylindrical arm, such as 50 mm, 60 mm, 70 mm, 80 mm, 90 mm, and 100 mm.

Point 3: Previous Concern # 5: the simulated radiation pattern is suspicious. Please provide the radiation pattern inside the Chamber including the real or phantom model, antenna mounted.

Point 4: “Figure 22, Antenna position on affected and normal feet”. The authors provided antennas on both feet (affected foot and normal foot). Therefore, It would be great if the authors can provide plots for S-parameters (S21) in LOS and/or NLOS for on-body locations (Concern # 8).

Reviewer 2 Report

I appreciate the authors' response and the revisions. While the paper has improved there are still major changes needed in the paper.

-The English grammar needs to be improved. A spell and grammar checker may be used to improve the English. For example "polyimide" is misspelled. 

-The antenna was not designed for placement on the human body, all the initial design results are for free space. Therefore, to keep the paper cohesive and focused on the EMC method, Section 3 and all the information about the placement on the body should be removed. The information is only based on simulation and does not provide sufficient information. It only takes the reader's attention away from the paper's main message.

-EMC method is still not clear to me. The first few steps to design the antenna structure are given, without any reasoning of why pieces are added or removed. Then a table and a figure are given. What is the equation to calculate EMC? Is Figure 10 show the same values as Table 1? Why both are of them needed? Why the frequencies in Table 1 are random? 

-Similarly, Figure 14 and Table 2 give EMC for the 2nd antenna, but not clear how the values are obtained. What is expected? How this value caused any change in the antenna structure?

-If the purpose of EMC is to increase gain, the gain has to be measured (before and after) to verify the method. If the authors can not do this in an anechoic chamber, they can measure the path loss between 2 known antennas and extract the gain.

-Authors should present and discuss the cross-pol vs co-pol levels. Most of the small monopole antennas suffer from high cross-pol, especially at the higher end of the band. The authors need to add co- and cross-pol pattern plots and they may remove the 3D pattern plots.

-Figure 8 does not provide any additional information and may be removed.

-Please remove this sentence: "... the antenna's radiation mainly depends on the acceleration of charge or variable currents."

-what does this mean: "next, the radiating patch is integrated with substrate and excitation", so that means there was no excitation or substrate up to that point?

-what is "xxx" in "polyimide film with a thickness of xxx mm"

-Considering the removal of performance on the body section, SAR values should be removed from the abstract, however, for the future please note that we mostly report the average 1g or 10g SAR, the peak SAR values do not provide the full picture. 

-For future work, if considering antennas as wearable ones, please include the human tissues such as a layered body model or a realistic body model as a part of the antenna design process.

Reviewer 3 Report

Authors had made an effort to improve the overall quality of the work. The reviewer has no furthered comments for this work. The current form of the work can be accepted for publication.
